# Polar Glycerolipids and Membrane Lipid Rafts

**DOI:** 10.3390/ijms25158325

**Published:** 2024-07-30

**Authors:** Anatoly Zhukov, Mikhail Vereshchagin

**Affiliations:** K.A. Timiryazev Institute of Plant Physiology, Russian Academy of Sciences, Botanicheskaya Street 35, Moscow 127276, Russia; zhukov_anatoly@list.ru

**Keywords:** lateral membrane heterogeneity, mechanisms of nanodomain formation, molecular dynamics, physicochemical properties of lipid bilayers, lipid phases Lo and Ld, glycerophospholipid groups with different unsaturation, lipid-lipid H-bonds, model biomembranes, computer modeling

## Abstract

Current understanding of the structure and functioning of biomembranes is impossible without determining the mechanism of formation of membrane lipid rafts. The formation of liquid-ordered and disordered phases (Lo and Ld) and lipid rafts in membranes and their simplified models is discussed. A new consideration of the processes of formation of lipid phases Lo and Ld and lipid rafts is proposed, taking into account the division of each of the glycerophospholipids into several groups. Generally accepted three-component schemes for modeling the membrane structure are critically considered. A four-component scheme is proposed, which is designed to more accurately assume the composition of lipids in the resulting Lo and Ld phases. The role of the polar head groups of phospholipids and, in particular, phosphatidylethanolamine is considered. The structure of membrane rafts and the possible absence of a clear boundary between the Lo and Ld phases are discussed.

## 1. Introduction

The cell membrane is probably the first achievement of evolution in the creation of life and remains one of the poorly understood structures of cells. Recent studies show that cell membranes are a complex matrix consisting of various specialized meso-domains with a specific structure, such as lipid rafts and protein-glycoprotein complexes. Modern understanding of the structure and functioning of biomembranes is impossible without determining the mechanism of formation of such rafts, their spatial and temporal parameters, and their existence under changing physiological conditions when they act as buffers of the physical properties of the membrane.

Cell membranes demonstrate a wide variety of lipids and proteins designed to perform the functions required by cells. It is known that the membranes in eukaryotic cells contain more than a thousand different types of lipids [1,2], including up to 40,000 individual lipids [3], but the functional meaning of this diversity remains unknown. The main structure of cell membranes is an amphipathic lipid bilayer, which surrounds cells and organelles and consists of two adjacent monolayers that form a two-dimensional “liquid” with the most important properties for living tissues. This bilayer is designed to perform a number of critical functions; in this way, it blocks the leakage of a number of hydrophilic compounds and retains membrane proteins [2]. 

To coordinate these functions, the membrane is able to separate its components in lateral directions. This ability is based on the dynamic immiscibility of different membrane regions and underlies the concept of membrane separation into lipid rafts [2]. Lipid rafts are oscillating nanoscale assemblies of sphingolipids, cholesterol (Chol), and proteins that form platforms capable of transmitting signals and transporting substances across membranes. This principle combines the self-assembly potential of sphingolipids and Chol with protein specificity for selective focusing of membrane bioactivity [2,4]. 

The most important milestones in the study of biomembranes can be considered the proposal of the liquid mosaic model by Singer and Nicholson in 1972 [5], the discovery of liquid-ordered (Lo) and liquid-disordered (Ld) lipid phases [6], and the discovery of lipid rafts in cellular membranes [7,8,9]. It became obvious that lipids can create non-covalently bound supramolecular structures with very specific physical properties [1]. Thus, cell membranes are a complex mosaic of domains containing various clusters of proteins and lipids [1,10]. What causes this segregation and what mechanisms are responsible for maintaining this topological organization under physiological conditions is a central problem for understanding the function of membranes [11].

The physical state of biological membranes is determined by the composition and structure of individual polar lipids (PL), free sterols and various proteins, and especially the head groups of these lipids, the degree of unsaturation, and the chain length of bound fatty acids (FA). These factors play an important role in regulating stress responses in various plant organisms. In plant cells, membranes can be the primary sensor for changes in environmental parameters. These changes are perceived by membrane sensory proteins that carry a signal from an external stimulus into the cell, triggering regulatory processes related to signal transmission and the expression of certain groups of genes necessary for the body to adapt to new conditions [12].

Recently, most of the works on membranology have been devoted to the study of the plasma membrane (PM) as a presumably classic model for organization of other biomembranes. In these studies, as a rule, the role of free sterols (for example, Chol), sphingophospholipids (for example, sphingomyelin (SM)), and proteins was considered, while the role of glycerophospholipids (GPhLs) was still insufficiently studied. First of all, attention was paid to completely unsaturated GPhLs, which form the Ld phase, as well as those that are unsaturated in only one sn-position of glycerol, since they are most common in PM [13]. Currently, the number of studies aimed at a variety of membranes of all organelles of eukaryotic cells is increasing. These biological membranes, in addition to proteins and sterols, consist not only of such sphingophospholipids as SM but also include various types of GPhLs. Moreover, it is important to keep in mind that each of the GPhL species consists of three groups of lipids with different degrees of unsaturation, including either two saturated FA (group 0), or unsaturated only in the sn-2 position (group 1), or unsaturated in both the sn-1 and sn-2 positions glycerol (group 2). Now numerous studies with model mixtures of lipids use GPhLs with different degrees of unsaturation (groups 0, 1, and 2), given that they can also be components of domains or lipid rafts [13,14,15]. The role of other glycerolipids—such as glyceroglycolipids—in the formation of lipid domains in membranes also requires further study.

The overwhelming role of Chol and SM (other name—ceramid phosphorylcholine) is mainly true when it comes to PMs of animal tissues. However, in most eukaryotes, in plants, SM is found only in trace amounts and is replaced by sphingoglycolipids, and Chol is replaced by other sterols—β-sitosterol, stigmasterol, and campesterol. As for Chol, as well as free sterols and glycosides of plant sterols, we refer to them not as lipids but as an independent group of compounds [12]. The presence of bound sterols in cell membranes, which include sterol esters and sterol glycoside esters, has not been reliably determined. 

## 2. Lateral Inhomogeneity of Membranes

Bilayer lipid membranes are formed as a result of self-assembly processes. Molecular interactions within the bilayer and with the environment create a unique profile of transverse pressure within the bilayer and provide a set of physical mechanisms for the formation of lipid domains and laterally differentiated regions in the membrane plane [1]. One of the first proofs of the existence of lateral domains is the presence of morphologically different structures called caveolae in PM mammalian cells [16].

Cell membranes are structured and complex fluids that are highly dynamic systems and constantly cross-interact with the cytoskeleton or extracellular matrix [3,11]. It is known that the cortical cytoskeleton containing actin and transmembrane proteins (TM proteins) can organize lipids into short-lived nanoscale assemblies, which under certain conditions can be assembled into larger domains [17].

The organization of eukaryotic cells relies on dynamic, continuously reconstructed molecular interactions. These interactions ensure constant renewal of the cellular composition and give plasticity to the cellular architecture. Lateral diffusion barriers play an important role in the formation and maintenance of specialized architecture in cell membranes. These barriers help cell membranes divide laterally into separate domains. Such barriers are found in the continuous membranes of the PM and endoplasmic reticulum (ER) and limit the lateral diffusion of membrane-bound proteins. As a result, each of these proteins diffuses freely in one of the membrane domains but cannot transfer to other regions of the membrane [18].

It is assumed that the division of the membrane into domains can serve as one of its main distinguishing features and is universal. Membrane domains can be defined as separate regions in which the local composition, lateral organization, and dynamics of their components differ from the properties of the averaged membrane. This specificity is predetermined by the preferred intermolecular intramembrane interactions of lipid–protein, lipid–lipid, and protein–protein, as well as associations with structures peripheral to the membrane, such as the cortical cytoskeleton and cell wall in the case of PMs [19]. 

There are various levels of structural asymmetry and heterogeneity in the membrane, including the distribution of lipids between monolayers, lateral distribution and organization of lipids, and interactions between lipid and non-lipid components. Lipid and sterol molecules do not form a single phase in the membrane but consist of at least two phases that coexist in dynamic equilibrium under physiologically acceptable conditions—liquid-ordered and liquid-disordered. These phases are characterized by varying degrees of ordering of lipid hydrophobic chains, membrane thickness, elastic modules, and other parameters. This ensures a sufficient variety of environments necessary for the proper functioning of a large number of membrane proteins that co-exist in the same cell to maintain its homeostasis [3].

Repulsive forces can be very important in the lateral organization of membranes, because repulsive interactions, such as lattice deformation, can have a much larger range of action than attractive forces, which usually act at the level of a neighboring molecule. There are several types of repulsive interactions that can play an important role in the lateral organization of membranes. These include various modes of steric deformation, Coulomb repulsion, and dipole interactions [16].

The structure, dynamics, and stability of the lipid bilayer are controlled by thermodynamic forces leading to the formation of tension-free bilayers with a noticeable lateral pressure profile and bending stress instability. These properties can manifest themselves locally or globally in the form of morphological changes leading to the formation of non-lamellar and curved structures. The molecular shape of various types of membrane lipids through the curvature stress serves as a regulator of the connection of these lipids with proteins [20]. 

Typical cell membranes can contain many different lipids, asymmetrically distributed between two monolayers, and are saturated with proteins that occupy up to 30% of its total volume. Lipids and proteins can co-form nanodomains (ND) that are important for many cellular processes, such as membrane fusion, protein transport, and signal transmission [21]. The lipid envelope of membrane proteins includes a belt of ring lipids that define the intimate environment of membrane proteins and non-ring lipids that interact with them in a more specific way. Only a small fraction of lipids in the membrane are not affected by any interaction with proteins or polysaccharides [3].

The hydrophobic correspondence between the TM proteins domains and the two lipid layers is probably the physical mechanism by which the membranes self-organize in lateral directions. Inconsistencies of TM proteins of different lengths with bilayers of different thicknesses are experimentally studied at the molecular level. Both model membranes and computer simulations have shown that Chol critically restricts structural adaptations at the protein–lipid interface under hydrophobic mismatch. These constraints are converted into a sorting potential and lead to selective lateral segregation of proteins and lipids according to the width of their hydrophobic parts [22].

It is obvious that the structuring of rafts and many of their structural and functional properties is carried out through the cortical actin cytoskeleton [23]. Issues related to abnormal and anisotropic diffusion in membranes are usually considered together with the problem of formation of lipid rafts. At the same time, these processes overlap with interactions with the expanded cytoskeletal network [24]. It is assumed that the membrane-bound cytoskeleton may play a role in regulating the size of rafts and thus provide an answer to the obvious discrepancy between the size of lipid microdomains in cell membranes and in model lipid mixtures. While lipid-driven segregation can be combined with additional actin cytoskeleton-based processes for the spatial organization of the dynamic status of membrane proteins in animal cells, similar actin-dependent mechanisms have not yet been fully described in plant cells [25]. 

The well-known “mattress model” shows that embedding a rigid helical TM protein in a liquid bilayer leads to local adaptation of lipids to mismatch, and the effect of their hydrophobic surface is minimized. Adaptive folding and straightening of lipids can also be accompanied by protein tilt. Further studies have shown that TM proteins admit a moderate mismatch with the bilayer.

It is clear that sphingolipids and Chol can self-associate into micron phases in model membranes, and these lipids are involved in the formation of highly dynamic nanoscale heterogeneity in the PMs of living cells. Functional heterogeneity of rafts occurs due to the principles of phase segregation caused by lipids in combination with the chemical features of their molecules [26]. 

## 3. Membrane Lipid Rafts

### 3.1. The Lipid Raft Hypothesis

Cell membranes are composed of phospho- and glycolipids, Chol, and membrane proteins that have a heterogeneous distribution. The study of membrane heterogeneity has been further developed with the improvement of the “lipid raft” model by Simons and Sampaio [2,4,15,27]. This model is based on the concept that certain lipids have a different biophysical tendency to bind to each other and in its simplest form assumes the presence of lateral inhomogeneities in the PM, which arise due to a denser packing of Chol with saturated sphingolipids or GPhLs (group 0), in contrast to disordered regions containing unsaturated GPhLs (group 1 and 2), which, in addition, can include not only mono- but also di-, tri-, and polyunsaturated FA in their molecules [28].

The lipid raft hypothesis is based on the occurrence of temporary predominant associations of individual lipids in certain areas of the lipid bilayer. Rafts can grow, cluster, contract, or disintegrate due to self-diffusion of their constituent lipids in accordance with functional requirements regulated by rather subtle changes in the activity of membrane ordering or disordering compounds [7,29]. The meaning of the word “raft” remained uncertain for a long time, until, in 2006, a “consensus” definition was formulated that membrane rafts are small (10–200 nm), heterogeneous, highly dynamic, sphingolipid- and sterol-rich (primarily Chol) domains that separate cellular processes. The concept of lipid raft perception is based on the idea that very small pre-formed domains are grouped under certain stimuli into functional microdomains. Small rafts (ND and ultra-ND) are sometimes stabilized, forming larger platforms due to protein–protein and protein–lipid interactions. According to this model, preferential interactions between Chol and sphingolipids lead to the formation of the Lo phase, in which lipids are more densely packed relative to the surrounding [2]. Membrane lipid rafts are characterized as microdomains, according to another version, nanoscopic assemblies of the lipid bilayer of cell membranes, in which regions with a denser packing of lipid molecules (Lo phase) are formed around certain proteins, alternating with less organized and more fluid bilayer sections representing the Ld phase [30]. Rafts isolated from different membranes can differ both in their biochemical composition and in their functions [31]. Thus, rafts do not always contain sphingolipids [32]. Rafts in model membranes independent of Chol have also been reported [33]. 

“Rafts” floating in the lipid “sea” of the membrane can rather be compared to “icebergs,” since they penetrate deep into the interior and do not float on the surface and consist of the same material as the “sea.” Lipid rafts, in all likelihood, can be of three types, similar to integral, surface, and TM proteins: (a) occupying part of one layer of the bilayer membrane; (b) the same, but with a larger monolayer thickness due to the inclusion of glycosylphosphatidylinositols (GPI) proteins on top of the membrane; and (c) occupying the bordering areas of both layers of the bilayer membrane (Figure 1). 

Lipid-protein microdomains called rafts are involved in many vital processes in the plant cell, such as lipid homeostasis, exo- and endocytosis, cell division, polarization, intracellular signaling, formation of membrane contacts, relationships with the cytoskeleton, etc. [34]. At the moment, rafts have been found only in the PM and tonoplast [35], but they may also be found in the membranes of many other organelles. 

Lipid rafts are usually enriched with saturated phospho- or glyco-sphingolipids, saturated (group 0) and/or only half-saturated (group 1) GPhLs, free sterols (Chol in the case of animal tissues), integral and TM proteins, surface membrane proteins anchored to GPI, and proteolipids containing C_14–16_-FA [28,36]. Rafts form much more ordered domains, moreover having a greater thickness than the surrounding parts of the membrane. Their formation is caused by lipid–lipid and lipid–protein interactions. The concept of a lipid raft is supported by a lot of experimental data—for example, tracking the diffusion of lipids or microscopy with super-resolution. Microdomains and ND in membranes are also observed in unicellular organisms and in prokaryotic cells [32].

The concept of lipid rafts is both widespread and highly controversial. Living membranes are extremely complex and diverse and can contain many different types of lipid rafts that can only exist on certain spatial and temporal scales [37]. The main difficulty in studying rafts is the lack of direct microscopic detection of these regions in the undisturbed state of a living cell, as well as in determining their size, life span, and frequency of occurrence in bilayers [12].

The term lipid rafts usually refers to relatively large and long-lived domains in biological membranes. It is assumed that they accumulate certain membrane proteins (often anchored by saturated alkyl chains), which can, for example, significantly enhance signaling cascades or significantly affect many other functions. The protein component of rafts is represented by aquaporins, ATPases, caveolins, G-proteins, protein kinases, and other proteins, and it can differ significantly [28]. The lipid component is more conserved than the protein component [28,31]. Specific TM proteins are interpolated into the lipid raft bilayer along with GPI-bound receptor proteins located on the exoplasmic side (Figure 1B) and FA-bound effector proteins on the opposite side (Figure 1F) [38]. 

The traditional method of studying lipid rafts and their association with certain membrane proteins is based on the fact that detergent-resistant membrane regions exist. One of the first indications of the existence of rafts was the detection of specialized lipid fractions consisting mainly of SM, Chol, glycosphingolipids, and saturated phospholipids, which are highly resistant to extraction with Triton-X 100 detergent at 4 °C; they were called detergent-resistant microdomains (DRM). Co-purification and isolation of such complexes were considered as evidence for the presence of domains in the membranes [39]. Such detergent-insoluble complexes can form GPI-bound proteins, lipid-bound non-receptor tyrosine kinases, sphingoglycolipids, and Chol. However, detergents can cause false phase separation, affect the separation of membrane proteins in a particular phase, and change the composition of lipids [40]. Therefore, the fact of DRM isolation cannot yet serve as proof of the existence of the same domains in living cells. Later, DRM-like domains were isolated by centrifugation in a sucrose gradient with a low-density fraction [41], and the use of Förster (or fluorescence) resonance energy transfer (FRET) and multiphoton microscopy methods made it possible to detect microdomains in living cells [42].

### 3.2. Lipid Rafts Formation

Model membrane systems are widely used to enhance the understanding of the chemical and physical characteristics of segregated lipidic phase domains. The deposited mono- and bilayers of binary phosphatidylcholines (PC) mixtures have segregated phase domains ranging in size from a few microns to tens of nanometers. Micrometer-sized domain structures were found both in mixed bilayers of various phospholipids and in mixtures of phospholipids with sphingophospho- and sphingoglycolipids [43]. It has already been mentioned that some membrane structures that differ in their biochemical characteristics are located in different zones of the sucrose gradient. The results of the analysis made it possible to identify three zones in the sucrose gradient after fractionation of the vacuolar membrane and two zones in the PM, where membrane structures were represented, which can be defined as lipid rafts by their lipid composition [31].

A model of possible organization of membrane rafts is shown (Figure 1A–C). In this model, both sphingophospho- and sphingoglycolipids, as well as GPhLs, associate laterally with each other, probably through weak interactions between their head groups. In sphingoglycolipids, such carbohydrate head groups occupy large regions in the plane of the exoplasmic monolayer, in contrast to their predominantly saturated lipid-hydrocarbon chains. Any voids between the associated sphingolipids are filled with Chol molecules that function as spacers [7]. Densely packed clusters of sphingolipids and/or saturated phospholipids and Chol behave as assemblies inside an exoplasmic monolayer (Lo phase), where the intermediate regions of the liquid are occupied by unsaturated GPhL groups 2 or 1 (Ld phase). Sphingoglycolipids usually contain long chains of FA that are amide-bound to the sphingosine base and can connect to the cytoplasmic layer of this bilayer. Chol also functions as a spacer in the cytoplasmic monolayer, filling the voids formed by intertwining FA chains. The nature of GPhL microdomains occupying the cytoplasmic side of the membrane is less well known, but they probably also contain mainly saturated FA chains. The Ld zone is expected to include unsaturated GPhL groups 1 and 2, as well as Chol.

It is assumed that rafts are formed by self-association of glycosphingolipids due to their long and mostly saturated hydrocarbon chains. The interaction between these lipids can be enhanced by hydrogen bonds between their head groups. Voids between hydrocarbon chains formed by rather bulky head groups are filled with Chol, which can also participate in the formation of hydrogen bonds with sphingolipids [44]. 

Membrane microdomains are dynamic assemblies that result from the lateral interaction of lipids and proteins. Chol and sphingolipids are thought to play a key role in mediating lipid-based interactions. It can be assumed that the formation of ND is influenced by the chemical, electrostatic, and geometric properties of lipids. In addition, the influence of curvature, asymmetry, and ions on membrane NDs is shown to be a very important aspect that can also modulate lipid NDs in cell membranes [45]. However, this concept remains controversial, as the study of microdomains in living cells is difficult due to the very dynamic and complex environment. Association and dissociation of membrane microdomains occur on an extremely fast time scale. In addition, only a few methods can detect lipid segregation in ND and especially in ultra-ND with a radius of less than 5 nm; one of them is the FRET method [46]. 

The temperature range at which ultra-ND is formed strongly depends on the type of lipids. Their formation in PM vesicles occurs more frequently in the presence of SM than dipalmitoyl phosphatidylcholines (DPPC). In general, vesicles that were closest to PM mammalian in lipid composition (SM/palmitoleoyl phosphatidylcholines (POPC)/at least 30% Chol) formed ultra-ND in a very wide temperature range. Experiments with the de-quenching of fluorescently labeled lipids have shown that domains with a size comparable to the Foerster radius (~5 nm) begin to form precisely at T ≤ 37 °C [3]. Nanoscale lipid assemblies are induced by SM-SM and SM-Chol contacts via intermolecular hydrogen bonds and Van der Waals interactions [34]. It is assumed that natural mammalian lipids tend to form ultra-ND compared to larger domains. Such NDs have been shown to be crucial for the proper functioning of membranes in cells [47]. 

It is known that the domain structure is created by differences in the physicochemical properties of membrane lipids, such as the phase transition temperature, intermolecular hydrogen bonds, and ionic functional groups. Domains are also created by specific interactions between various membrane lipids with the formation of stoichiometric complexes, which are often transformed into ordered structures, which are membrane rafts [48]. 

Thus, SM and Chol are typical components of lipid rafts. SM, forming rafts, acts as a key lipid molecule in the membranes of animal tissues. It is believed that specific proteins and lipids found in rafts form dynamically ordered domains of submicrometer size. Lipid microdomains can selectively include or exclude membrane proteins [49]. A variety of cellular processes such as cell adhesion, signaling, and microbial invasion can be mediated by lipid rafts, in which the dynamic balance of formation and decay is considered a key factor for many cellular functions [50].

The raft structure in PMs can be represented as an asymmetric bilayer with sphingolipids located on the exoplasmic monolayer and GPhLs, such as phosphatidylethanolamine (PE) and phosphatidylserine (PS), located in the cytoplasmic layer, with a Chol distribution between the two layers. The rafts are surrounded by liquid bilayers consisting mainly of unsaturated GPhL groups 1 and 2. It is likely that the two sides of the membrane are connected by interweaving long N-acyl chains of sphingolipids of the exoplasmic layer with Chol intercalated with GPhLs of the cytoplasmic monolayer [38].

There is a demand for model systems with reduced complexity that still sufficiently mimic PMs [50]. The formation of discrete microdomains (rafts) alternating in the continuous Ld phase is also observed in model membranes. Sphingolipids (primarily SM), Chol, and saturated GPhL group 0 can easily associate to form domains. The Lo state is crucial for the raft’s existence. Sphingolipids contain mostly long saturated acyl chains that allow them to pack tightly together. This property of sphingolipids makes their melting point (Tm) higher compared to the mixture of natural GPhLs, which usually contains more unsaturated acyl chains [41]. 

An important contribution to our understanding of the structure of biomembranes is the detection in bilayers containing high proportions of Chol of the coexistence of two liquid lipid phases Lo and Ld—a phenomenon not observed in mixtures of phospholipids in the absence of sterols [39]. Sometimes, due to the supposed high concentration of sterols in rafts, they are considered to be in the Lo phase. It is assumed that ND can spontaneously form in the presence of Chol, which plays a crucial role in their stabilization by reducing the tension between the Lo and Ld domains [28].

Using large-scale full-atom molecular dynamics (MD) simulations performed on a model membrane (1050 lipid molecules), we observed the time evolution (6 microseconds) of raft-like domain consisting of palmitoyl sphingomyelin (PSM) and Chol and embedded in the surrounding non-raft-like region consisting of palmitoyl docosahexaenoyl-PC (PDPC, group 1; 1:1:1, mol.). It is noted that PDPC molecules in small numbers penetrate into the ordered raft-like domain, while some of the PSM and Chol molecules migrate to the disordered region that is not associated with the raft. In addition, a narrow (2 nm wide) interphase zone consisting of PDPC, PSM, and Chol is formed around the raft, which buffers a significant difference in the order [51]. In this experiment, the interphase zone consists of a mixture of lipids and Chol included in both phases (Lo and Ld), but it could presumably consist, in their presence, of lipids such as POPC, intermediate in unsaturation between PSM and PDPC, while the Ld phase is optimal for dioleoyl phosphatidylcholines (DOPC) (group 2).

Thus, it is possible that there are no clear boundaries between the Lo and Ld domains, but there is an intermediate zone (Figure 1D) [51]. If the Lo phase is dominated by DPPC and Chol, and the Ld phase is dominated by DOPC, then the intermediate zone may consist of POPC. This zone can be blurred, because POPC is also included in both adjacent phases (because there are a lot of them in the cell). It is possible that this is one of the reasons why it is so difficult to determine the size of rafts. 

Combinational (Raman) microscopy was used to directly visualize the distribution of diyne-SM in domains simulating rafts formed in triple monolayers of SM/DOPC/Chol. Raman images visualized an inhomogeneous diyne-SM distribution that showed noticeable differences even within the same ordered region. In particular, the central region of the lipid rafts was enriched with diyne-SM subdomains compared to their peripheral region. These results seem inconsistent with the generally accepted raft model, in which the raft and non-raft phases exhibit a clear two-phase separation. It is possible that gradual changes in the SM concentration occur in order to minimize the hydrophobic division. Oleosome membranes containing natural SM and its enantiomer were studied by the FRET method, showing that they form separate subdomains with sizes up to nanometers within Lo domains. Temporarily stable but mostly dynamic domains were observed in the 5–60 nm range by several methods, including interferometric scattering microscopy and stimulated radiation depletion microscopy [34].

### 3.3. The Size of Lipid Rafts

When the domains have a diameter of less than 100 nm, they are called ND [52]. Such NDs can be temporarily grouped to form large rafts, including many proteins necessary for the typical functions of these rafts. In living cells, NDs are reliably identified only due to the advent of super-resolution microscopy techniques. ND is stabilized by low linear stresses and high dipole–dipole interactions and crowding pressures [29,52]. Often, the ND size is less than 10 nm, which corresponds to the size of a single protein complex [53]. In addition, the ND lifetime is typically less than a few nanoseconds [47,53]. The most studied are long-lived domains (rafts) of relatively large size, exceeding 100 nm; in practice, they are sometimes also called ND. Terminology ND is also sometimes used for large domains (up to 200 and even 300 nm) [53].

A three-level hierarchical architecture is proposed for all mesoscale (2–300 nm) domains represented in the fundamental functional organization of PMs. All these membrane domains are proposed to be divided into three groups or levels. The most basic domain in the hierarchy is the “membrane compartment” with a size of 40–300 nm, which is created by dividing the entire PM, due to its interaction with the membrane framework containing actin and TM proteins attached to the membrane framework fence. The second in the hierarchy is the “lipid raft domain” with a size of 2–20 nm, created by various levels of molecular affinity and immiscibility, including both lipids and proteins, as well as joint condensations within the PM, which can be modulated by stimulation-induced dimers and oligomers of raft-associated receptors. The third domain includes dimers, oligomers, and larger complexes of membrane-bound and integral membrane proteins (3–10 nm). Protein complexes are considered as one of the meso-domains that are formed without any significant participation of lipids [54].

Since the second level in the proposed scheme includes only those rafts that are ND, and given that many researchers consider rafts and domains with a size of 20–200 nm [47,53], we can suggest dividing the first level in this scheme into two groups with the size of domains of membrane compartments of 200–400 nm [17] and domains of lipid rafts with diameters of 20–200 nm. Thus, both ND (2–20 nm) and larger domains (20–200 nm) can be called lipid rafts. 

Lipid rafts are too small to be detected by methods such as fluorescence microscopy [37]. Direct evidence for the presence of rafts in vivo is mainly based on monitoring the movement of membrane proteins or on differential separation of fluorescent probes in membrane media. It is quite difficult to measure the physical quantities of rafts, such as the exact lipid composition, characteristic size, and lifetime, when using live cells. The co-existence of the Lo and Ld phases is widely recognized in model membranes. It is considered that to obtain them, it is necessary to take the minimum three-component SM/Chol/DOPC mixture. Here, the Ld phase can be formed by unsaturated PC, while the Lo phase is created by a mixture SM и of SM and Chol. Lipid rafts in biological membranes represent ND of various compositions, rather than large-scale phase separation [37]. Individual lipids can enter and exit rafts [7], which explains why SM and Chol clusters are difficult to detect spectroscopically. 

The size of a lipid domain is one of the physical properties that can have an important impact on domain function. It is known that for some lipid compositions, ordered domains exist in the form of large-scale micron-sized phases, while in other cases, they exist in the form of ultra-ND with a size of only one or several nanometers [47]. It was found that the hydrophobic mismatch between Lo and Ld domains with different thicknesses contributes to the functioning of these domains [46]. A simple and affordable method for measuring domain sizes below the optical resolution (200 nm) is proposed. The ND size was measured for two mixtures, distearoyl PC (DSPC)/POPC/Chol and SM/POPC/Chol, by combining FRET data with Monte Carlo analysis. The ND radii were ~9 and 5 nm for these two mixtures, respectively. The existence of ND in larger domains of the Lo phase was detected by experiments with fluorescence lifetime and 2H NMR. It is shown that the ND radius can decrease from ≥15 to <4 nm as the temperature increases from 10 to 45 °C [47].

Membrane microdomains can be detected by atomic force microscopy (AFM) using differences in height between lipids existing in different phase regimes, including rafts or Lo phases, which can exceed the surrounding Ld phase by 0.5–1 nm in model membranes consisting of DOPC/SM/Chol. ND research requires the use of both ACM and super-resolution fluorescence microscopy as the main tools [43,52]. 

### 3.4. The Role of Lipid Proteins in the Formation of Cholesterol-Rich Domains

Rafts regulate the sorting of associated proteins. In this case, the heterogeneous distribution of membrane proteins occurs due to the spontaneous grouping of lipids with the formation of domains with different properties and, consequently, different media for certain membrane proteins. Since sphingolipids, which can be involved in cell signal transmission, are usually concentrated in rafts, it can be assumed that rafts can also participate in intercellular recognition and signals transmission [32,39]. 

Lipid-bound proteins are organized into Chol-dependent domains or submicron-sized rafts. These domains probably have a diameter of less than 70 nm and are destroyed when the cellular Chol is removed. GPI-anchored proteins are a diverse set of membrane-bound exoplasmic eukaryotic proteins that appear to be diffusely distributed over the surface of these membranes. The latter contradicts the existence of rafts, but it was the functional properties of GPI-anchored proteins that provided the first and most convincing arguments in favor of lateral segregation of these proteins [55]. When proteins attached to GPI or TM proteins are bound to the raft, their diffusion becomes independent of the type of membrane anchor and is significantly reduced compared to that of other proteins. Lack of Chol accelerates the diffusion of raft-bound proteins. However, GPI-bound proteins attached to the raft were never observed to separate from the raft within the first 10 min. The measurements are consistent with the fact that the lipid rafts are stabilized Chol complexes with a size of 26 ± 13 nm, diffusing as a single unit for at least 5–10 min [44].

While lipids are known as a physical barrier, membrane proteins have been shown to determine the physical and chemical properties of membranes. Thus, complex interactions between membrane proteins and lipids can cause curvature of biological membranes, which leads to specific physiological functions [56]. Interactions between proteins and lipids underlie almost all membrane processes. Membrane proteins are known to bind non-covalently to lipids and sterols. A general analysis of Chol-binding proteins revealed 250 such proteins, including various enzymes, channels, and receptors that bind it via their TM proteins domains [3]. Membrane proteins are affected by such general properties of lipids as their packaging, lateral pressure profile in the membrane, internal curvature of lipids, bilayer thickness, and electrostatic properties [57]. 

It is recognized that the function of internal membrane proteins and the local lipid environment, characterized as membrane microdomains, can be strongly interdependent [50]. Thus, the function of many of these proteins requires a conformational transition, which is often strongly influenced by the molecular composition of the bilayer in which the protein is embedded. Microdomains of membrane rafts have the ability to influence the spatiotemporal organization of protein complexes, thereby allowing the regulation of cellular processes [47]. Lipids can affect the structure of membrane proteins by affecting the conformation of their backbone, the angles of inclination and rotation of their TM proteins segments, or the orientation of their side chains. Conversely, membrane proteins can stretch or disrupt the chains of surrounding lipids, promote inter-layer movement of lipids, influence their lateral organization, induce the formation of various macroscopic phases, or promote processes such as membrane fusion or division [57].

The formation of protein nanoclusters anchored to GPI (~4 molecules or even less) is an active process involving both actin and myosin, and these nanoclusters are distributed non-randomly into larger domains up to 450 nm in size [17]. When proteins attached to GPI and TM proteins are bound to the raft, their diffusion becomes independent of the type of membrane anchor and is significantly reduced compared to that of non-TM proteins. Lack of Chol accelerates the diffusion of raft-bound proteins [44].

It is known that proteins can change the dynamics of lipids around them, and there are protein-lipid complexes containing hundreds of lipid molecules. It is possible that these are the smallest functional units in cell membranes [58]. The stoichiometry of the lipid–protein interaction is important not only for structural reasons, but also because the first lipid envelope is responsible for sealing the protein in the membrane and, therefore, maintaining the permeability barriers in the cell. Studies of integral proteins by the spin-labeled electron resonance (ESR) method have shown that a constant amount of lipids is limited in motion by direct interaction with each protein, regardless of the total lipid content in the membrane. First-shell lipids are those that solvate the protein in the membrane and provide an interface with the liquid lipid medium, which is necessary for the functioning of many membrane enzymes. The first envelope or ring of lipids may also represent the minimum amount of lipids required to maintain TM proteins activity; these lipids are known as annular lipids [12]. Palmitoylation has been shown to regulate raft association for most of the integral proteins that make up the raft. Palmitoylation of cysteine is the only lipid modification that is reversibly regulated. It is possible that palmitoylation may be a dynamic mechanism for binding rafts to TM proteins [59].

One of the factors influencing the formation of Chol-rich domains in membranes is the uneven lateral distribution of proteins. There are several structural features of the protein that lead it to preferentially associate with Chol-rich domains. Thus, certain protein segments can directly bind Chol. There are proteins that preferentially bind to Chol-containing regions of the membrane, and, therefore, they directly stabilize these domains. Most of the proteins in PMs are bound to raft domains, and they are distributed between the Lo and Ld phases. It is possible that some proteins are sometimes excluded from the Chol-rich regions of the membrane, because their presence leads to denser packing of lipids [60].

Potassium channels are known to directly bind negatively charged lipids such as phosphatidylinositol phosphates. The activity of these channels is proportional to the concentration of phosphatidylinositol phosphates. It was found that channels reduced in a bilayer with cationic lipids lacking the phosphodiester group lose their characteristic voltage-dependent behavior. When anionic phospholipids are added to the bilayer, the channels restore their activity, which is controlled by this voltage. In this case, positively charged arginine side chains form hydrogen bonds with negatively charged phosphate groups of lipids [61].

## 4. Liquid Ordered and Disordered Lipid Phases

### 4.1. Formation of Lo and Ld Phases in Synthetic Membranes and Giant Single-Layer Vesicles

Saturated and unsaturated phospholipids and Chol are key components of biological membranes. Since sphingolipids have predominantly saturated chains and natural PCs are predominantly unsaturated, the composition of the outer monolayer of the membrane can be roughly represented as a mixture of Chol, saturated lipids, and unsaturated lipids. It is known that Chol is separated together with saturated lipids in the Lo phase, while unsaturated lipids are grouped in the Ld phase. In model membranes, the domains under consideration are relatively large and, therefore, can be visualized quite easily [58]. 

In addition to the Lo and Ld phases, a solid-ordered or gel phase can also exist in artificial membranes under certain conditions. This phase occurs in hydrated pure lipids. The gel phase contains lipids that have tightly packed acyl chains parallel to each other. In this phase, the lipids condense strongly and are essentially immobile. The gel phase in artificial membranes can co-exist with the Ld phase, which has a loose and free side packing of lipids, curved acyl chains with high mobility, rotating around an axis perpendicular to the membrane surface, and rapid lateral diffusion. There is an intermediate phase between the gel phase and the Ld phase, which is the Lo phase. A thermally stable gel phase induced by homophilic interaction of cerebrosides (Cer) with each other was found in the domains formed in the Cer/Chol/POPC mixture [51]. Lipids capable of forming the gel phase are high-melting SM, DPPC, and DSPC [62]. SM is usually used in the form of PSM [51].

As a result of work on model membrane systems, the membrane was presented as divided into lipid fractions in the form of Ld and Lo domains. The concept of the Lo phase was proposed to describe the physical state of lipids in lipid rafts. The Lo phase is explained as an intermediate state between the gel and liquid states, in which the acyl chains are still stretched (i.e., ordered) but do not have a hexagonal lateral arrangement of molecules and exhibit rapid diffusion. For pure lipids, both of these properties change simultaneously upon melting from the gel to the Ld phase, but this process can be changed in the presence of Chol to form the Lo phase [29]. The Lo phase is characterized by dense packing of ordered and elongated acyl chains but still has the same high lateral and rotational mobility as the liquid. As a consequence, the cross-sectional area per lipid decreases here [28]. Despite the fact that many properties of the Lo phase have been studied, its significance for rafts in cell membranes is still not sufficiently elucidated [5,6]. Thus, the size and physical properties of Lo domains formed in artificial membranes are very different from those found in PMs [63].

Thus, artificial membranes with a lipid composition similar to the outer PM monolayer are divided into Lo and Ld phases. Typical lipid compositions include low-melting GPhLs, high-melting GPhLs, or SM and Chol [63]. It is known that the rafts are thicker than the rest of the membrane; this is due to the presence of SM. Their molecules contain a long part of sphingosine and a long chain of saturated FA. Therefore, the rafts may have a greater thickness than the surrounding lipid matrix (Ld phase), which contains mainly medium-chain unsaturated phospholipids [41].

Since the qualitative composition of polar lipids (PLs) cell membranes is very diverse, the properties inherent in lipid rafts are usually studied in simplified model membranes with triple lipid compositions, where the Lo phases simulating rafts coexist with the non-raft Ld phase. It was found that in these membranes, the Lo phase consists mainly of saturated lipids with high Tm, such as DPPC and Chol, which have many properties of lipid rafts in cell membranes. It is shown that the Lo phase is crucial for the existence of the raft [41] and also that the rafts exist in the Lo phase [64]. Thus, the physical properties of lipid rafts appear to be markedly similar to those of the ordered Lo phase [50]. Unsaturated Gphls, which are part of the Ld phase, are loosely packed and separated from the Lo domain. It is believed that Chol and sphingolipids such as SM are necessary for the formation of lipid rafts, but the types of molecular interactions involved in this process, such as intermolecular hydrogen bonds, are poorly understood. An obvious question arises: why do natural lipid rafts prefer SM rather than saturated DPPC, even though both lipids form Lo phases in the presence of Chol? To answer this question, it is important to study the difference between membrane SM and DPPC molecules (Figure 2), especially from the point of view of the ordering effect of Chol on each of these lipids [64]. It was noted that the favorable effect of high Chol concentrations was greater in vesicles containing SM than in other lipid vesicles containing DPPC. The effect of SM was particularly large on the formation of ultra-ND compared to DPPC, which was not observed in the case of larger domains [46].

Thus, typical lipid compositions of artificial membranes that modulate the outer monolayer of PMs include three lipid components—a high-melting (high-Tm (melting point)) lipid (SM or, for example, DPPC, group 0), a low-melting (low-Tm) lipid, for example, POPC (group 1) or DOPC (group 2), and Chol—and they exhibit key properties associated with lipid rafts. These systems simply copy the composition of specific biological membranes and reproduce many complex phenomena, including the coexistence of the Lo and Ld phases. However, their compositional simplicity makes them suitable for studying composition- and temperature-dependent behavior (Table 1) [13].

In order to detect the formation of the Lo phase, while avoiding possible detergent artifacts, fluorescence quenching is used to study the phase behavior of model mixtures of PC, sphingolipids (in the form of SM). and Chol with differing unsaturation. Phase separation into Lo and Ld was found in binary mixtures (SM or DPPC)/POPC labeled with nitroxide (12SLPC). At the beginning of the experiment, the solid-ordered gel-like phase enriched in SM or DPPC co-existed with the 12SLPC Ld phase at 23 °C. At the same time, a fairly high concentration of SM or DPPC was required for the formation SM of the gel phase DPPC; otherwise, only a homogeneous liquid phase Ld was present. However, it is not denied that SM and DPPC operate in the same way. The addition of 33 mol.% Chol in the model membranes contributed to a new phase separation. In this case, the gel phase was eliminated in favor of the Lo phase, which then turned into the Ld phase with increasing temperature. Thus, in the Chol absence, no phase separation into Lo and Ld was observed. Additionally, the fact that phase separation occurs in membranes containing Chol confirms that the Lo and Ld phase domains co-exist. It is known that even double mixtures of palmitoyl-SM (PSM)/Chol, DPPC/Chol, POPC/Chol, and palmitoyl petroselinoyl-PC (PPetPC)/Chol can form the Lo phase (Table 1) [29,65]. Sphingolipids usually exist in a solid ordered (gel) phase, but the presence of Chol prevents them from forming a gel phase, converting it instead to the Lo state.

The separation of lipid phases is caused by interactions between Chol, sphingophospho- and sphingoglycolipids, and GPhL—saturated (group 0) and unsaturated (groups 1 and 2). It has already been mentioned that the presence of Chol in membranes can induce the Lo and Ld phases depending on its concentration. The Ld phase can change to the Lo phase if the Chol content exceeds 30 mol.%, even though its lipid composition (POPC) remains the same. Chol plays a unique role in the formation of the Lo phase, disrupting the translational order in the crystalline (gel) state of lipids and creating a high degree of ordering of acyl chains in the new phase, while maintaining a rather rapid lateral diffusion in it [28,66]. 

Micron-sized Lo phase domains can be seen by microscopy in giant single-layer vesicles (GUVs). GUVs are the most common of the model biomembrane systems and consist of either lipids or sterols, or additionally contain purified and reduced proteins. Depending on the lipid composition of GUVs, the Lo and Ld phases can co-exist in GUV membranes Lo и Ld. The disadvantage of GUVs is that they usually consist of only very few types of lipids and proteins, which makes it difficult to compare the results obtained in vitro with the actual situation in the cell [50]. The length scale of Lo domains in the two-phase domain strongly depends on their composition. DOPC/DPPC/Chol, perhaps the most studied model ternary mixture, forms micron-scale domains that can be observed by diffraction-limited fluorescence microscopy; the same domains are noted in GUVs. Replacing DOPC with POPC makes the composition more physiological and reduces the length scale of the Lo domains to tens of nanometers [15].

### 4.2. Lo and Ld Phases in Eukaryotic Membranes

PM studies were conducted mainly on animal tissues, and Chol and SM were considered to be the main components of lipid rafts [25]. It is assumed that the Ld phase consists of unsaturated GPhLs (groups 2 and 1) [64]. The study of model membranes is based either on simplified two-component models—DPPC or SM/unsaturated PC [65]—or three-component models consisting of Chol/SM/DPPC (or DSPC) [65], Chol/SM/POPC [47], or Chol/SM/DOPC [28,37]. The Chol/SM/POPC mixture has become a classic one and is considered minimal and sufficient for studying membranes (Table 1). In our opinion, this composition is insufficient, and the four Chol-components Chol/SM/POPC/DOPC mixture will be sufficient (Table 1). In this case, the Lo phase will include Chol, SM, and POPC, and Ld will include Chol, DOC, and POPC. POPC serves as a filler and stabilizer in both phases as the dominant component of natural membranes. Naturally, here SM is interchangeable with DPPC, and PC is interchangeable with phosphatidylethanolamines (PE) or other GPhLs. If we replace SM with DPPC, we can predict that the Lo phase will consist of DPPC + SM + Chol, Chol, the Ld zone of DOPC + Chol, and the intermediate zone of POPC + Chol (Figure 1D). Presumably, the intermediate zone is blurred, and POPC, as a major component of membranes, is also included in the Lo and Ld phases (Figure 1D). This may make it difficult to determine the size of the rafts. 

It should be noted that in the works of a number of authors, sometimes there is no clarity in the designations of lipids: if they refer to “sphingolipids,” then it is necessary to indicate which ones. sphingophospho- or sphingoglycolipids, and if they refer to “saturated phospholipids,” then emphasize that they are either SM (sphingophospholipids) or GPhLs [1]. Phosphatidylinositols (PIs) are among the numerous PL components of membranes; PIs bind surface proteins. It can be assumed that such high-polar lipids as phosphatidylserines (PS), sulfoquinovosyl diacylglycerols (SQDGs), and phytoglycolipids (PhGLs), which may contain elevated fractions of palmitic acid [12], as well as PI, can also form the Lo phase (Table 1).

The side diffusion coefficients depend on the packing of lipids and the ordering of acyl chains. It is shown that lateral diffusion is the same for all components, regardless of their molecular structure (including Chol), if they are located in the same domain or phase in the membrane. It is shown that Chol is apparently divided into Ld and Lo phases to approximately the same extent. Thus, Chol does not have a strong preference for any of these phases, i.e., it has similar interactions with all lipids [69]. Modeling of the MD triple mixture of Chol/DOPC/SM showed that Chol is preferably localized at the interface between regions enriched with saturated SM and unsaturated DOPC, with an entropically favorable packing of saturated acyl chains of SM on a smooth α-surface of Chol. At the same time, disordered unsaturated DOPC acyl chains are more easily packed with protruding methyl groups of the coarser β-surface of Chol (Figure 1E and Figure 3). In this simulation, small rams and linear ND were formed but did not increase in size during the simulation time [28].

Modeling of MD using all atoms of asymmetric multicomponent membrane bilayers in the mixed phase has shown that the immobilization of lipids with a long saturated acyl chain in one of the monolayers leads to interactions of the same lipids in the opposite layer. The membranes of living cells probably have a common mechanism for generating and stabilizing nanoscale Chol-dependent and actin-mediated lipid clusters [25,70]. In large-scale modeling of MD multicomponent lipid systems, it was observed that the stability of the Ld domain increases with increasing degree of unsaturation of lipids in this domain, and it was shown that an increase in the proportion of polyenex FA in these lipids contributes to the separation of the Lo and Ld phases. In the interaction between DPPC, POPC, and Chol, saturated lipids and Chol converge due to favorable interactions in the Lo phase, and both are repelled by unsaturated lipids in the Ld phase. Lipids with a higher degree of unsaturation have the expected even stronger repellent effect on Chol and saturated lipids. 

## 5. Contacts of Lipids with Sterols in Membranes

Free sterols, and especially Chol, which are representative in animal tissues, play a crucial role in the formation, existence, and functioning of biomembranes. Eukaryotic PM cells are known to contain about 20–50% sterols [71]. The absence of PSM and Chol in the raft-like membranes resulted in the formation of highly packed and rigid bilayers. [37]. It is known that the lack of Chol leads to the dissolution of model lipid rafts, while its addition provokes an increase in the size of SM-rich domains and, ultimately, leads to the formation of a single lipid phase similar to the raft. Chol is a key lipid in raft formation because it is highly miscible with SM and other phospholipids in bilayers. Chol is able to increase the fluidity of lipids forming the rigid gel phase (DPPC, SM) and, conversely, increase the order of liquid lipids (for example, POPC, DOPC), which leads to the formation of the Lo phase [62]. Colocalization of Chol with SM-rich domains SM was confirmed using the sterol-binding agent filipin. [33,72].

Chol has the formula C_27_H_46_O; in addition to it, in nature, there are also such sterols as ergosterol (C_28_H_44_O), stigmasterol (C_29_H_48_O), β-sitosterol (C_29_H_50_O), and others. Sterol molecules consist of a rigid hydrophobic core and a flexible hydrocarbon chain that is also hydrophobic, and the polar headgroup is the only OH-group that contacts the hydrophili head groups of PL (Figure 3). Sterols can be not only free but also bound. for example, esters of Chol, stigmasterol, β-sitosterol, and ergosterol with FA (C_12–20_). Plants also contain glycosides of sterols (Chol, sito-, and stigmasterols). From squalene, lanosterol is formed, and then its multi-stage conversion to Chol occurs. In this conversion, three methyl groups are lost, one double bond is broken down, and another is isomerized. Cholic acid can also be a precursor of Chol. The formulas of cholic acid, lanosterol, and Chol are quite similar to each other. Chol can then be transformed into the steroid hormones ergosterol, progesterone, testosterone and its acetate, and vitamin D_3_ and its esters, all of which also have a similar formula to Chol (Figure 3).

Each of the molecules of lipids, sterols, and proteins, in order to get into its place in the membrane, must combine its configuration with that of neighboring molecules. Sterol molecules are located in the lipid layer of the membrane parallel to the aliphatic chains of PL molecules. The presence of sterols in membranes reduces the mobility of FA and reduces the lateral diffusion of lipids and proteins and therefore can affect the function of membrane proteins [12]. Chol is particularly effective in eliminating empty space in the region of lipid acyl chains, which is due to the suppressed compressibility of this region and the increased bending stiffness of the membrane with increasing Chol concentration. However, it is assumed that the rate of lateral diffusion does not slow down more than two to three times when comparing the Ld phase with the Lo phase induced by Chol. Chol is known to significantly alter the lateral pressure profile of membranes. This is important because it was suggested that changes in the lateral pressure profiles are associated with changes in the structure and activity of membrane proteins [37].

It is known that only small amounts of Chol are required for a number of cellular functions, but for some reason, all eukaryotic PMs contain significantly more Chol (see above), which makes it the most common molecule in PMs. The question arises, what is so special about Chol, as well as other higher sterols? Interestingly, what can Chol do in membranes, in contrast, for example, to its biochemical predecessor lanosterol? It can be assumed that a small head group of Chol has an effect, but other sterols have the same head group, for example. lanosterol and ergosterol (see Figure 3), which are less common in eukaryotes. Chol has a cylindrical shape and more even walls without protrusions, unlike lanosterol, which has three uneven methyl groups on its *α*-surface. At the same time, plant sterols have such lateral processes in the form of methyl groups [37]. The question thus remains open (Figure 3). Do their esters with FAs, along with sterols, find application in membrane FAs? Figure 3 shows the formulas of Chol molecules and their esters, which show that in Chol, the head group is –OH, whereas in Chol esters, the head group is all the rest of the molecule, except –OH, which is replaced by the hydrophobic residue FA. Thus, in Chol esters, the main part of the molecule is closed on both sides by hydrophobic chains. At the same time, the polarity of these esters is much lower than that of the original Chol, and the hydrophobicity, respectively, is higher and comparable to triglycerides, which do not participate in the membrane structure.

Comparison of binary mixtures of Chol with GPhLs or SM showed that the way sterols interact with these types of phospholipids is different, and this is due to their special structure; however, in a series of computer models of hydrated bilayers of binary mixtures of Chol/stearoyl-SM, Chol/oleoyl-SM and Chol/POPC, it was found that the interaction energies between Chol and all three phospholipids do not differ significantly [73]. It has been shown that Chol strongly deforms the peptide–lipid interface at mismatch if the bilayer is thicker than the TM segment of the protein. In addition, it ensures the redistribution of lipids and TM proteins in accordance with the length of their hydrophobic part. It is possible that Chol reduces the adaptivity of lipids to mismatched proteins and, consequently, makes hydrophobic mismatch energetically more relevant [22].

What is the physical basis for the association of Chol and sphingolipids? Co-packaging of Chol with saturated acyl chains of sphingolipids is entropically more favorable than with unsaturated chains. Dipole interactions between sphingolipids and possible hydrogen bonds between the -OH group of Chol and the amide group of sphingolipids and ceramides may also contribute to a favorable association of Chol with sphingolipids. According to the “umbrella model,” Chol is distributed in areas of the membrane with highly hydrated large head groups, such as in sphingolipids, where sterol rings can be more effectively protected from the aquatic environment, thereby providing energetically favorable interactions. According to another scheme, Chol forms reversible condensed complexes of a certain stoichiometry with both sphingolipids and GPhLs of group 0 [64]. When comparing the physical properties of the stearoyl-SM enantiomer (ent-SM) with natural stearoyl SM, it was shown that PSM, which has a natural stereochemistry, showed higher miscibility with SSM bilayers than with ent-SSM bilayers, which indicates that SM homophilic interactions occur in a stereoselective manner [72].

In the phospholipid membranes of plant cells, glycosphingolipids form lipid rafts in which they are tightly packed with sterols. In these rafts, glycosphingolipids associate laterally with each other, probably through weak interactions between carbohydrate head groups. Any voids between the associated sphingolipids are filled with Chol molecules that function as spacers [7]. Cerebrosides (Cer; also known as glycosylceramides) are glycosphingolipids, consisting of a head group in the form of glucose bound to the ceramide backbone (Figure 4), and are involved in signaling to plants and are necessary for the viability of cells and the entire plant. The structural relationship of Cer-sitosterol in membranes with palmitoyl linoleoyl-PC (PLPC) was studied using Langmuir films, simulations in silicone, and neutron reflectometry. It has been shown that there is a strong direct interaction between these two molecules, which controls their lateral and transverse distribution within each monolayer of membranes [74].

While animal and fungal membranes each contain one major class of sterols, Chol and ergosterol, respectively, plant membranes include several sterols, such as stigmasterol, β-sitosterol, and campesterol. β-sitosterol has been shown to give phospholipid membranes a more ordered structure, similar to that of animal tissues containing Chol. MD modeling has shown that clusters of Cer and sitosterol are formed in the membrane with dilinoleoyl PC (DLPC), which indicates the tendency of these molecules to aggregate in domains separated from unsaturated GPhLs. Moreover, it was shown that model membranes imitating plant PM are less sensitive to temperature than animal membranes, which suggests that components such as sitosterol and Cer are produced to expand the temperature range in which biological processes associated with membranes can occur [74]. 

It has already been mentioned that the Ld-phase regions of the membrane are Ld generally thinner and less mechanically stable than their harder counterparts, which, on the other hand, do not provide rapid diffusion. First of all, the efficiency of Chol is very important for ordering the liquid phase of membranes and providing it with low passive permeability and increased mechanical strength [62]. The Chol molecule tends to stabilize the membrane by arranging the lipid acyl chains in the liquid portion of the membranes due to its rigid steroid structure and *α*-surface (Figure 3), which is molecularly smooth. In harder areas of the membrane, Chol has the opposite effect, limiting the packing density. Thus, when it comes to ordering, Chol molecules prefer solid phases, while in the case of packaging, Chol prefers the liquid phase. This duality in the affinity of Chol for the liquid and solid phases of the lipid bilayer allows us to confirm once again that the presence of large amounts of Chol induces a new mesophase, which is the Lo phase. The Lo phase is a primary phase in membranology, and it has been shown that no molecules other than higher sterols can stabilize this type of phase. Thus, Chol is unique in terms of its most similar ordering properties [75]. This also applies to other higher sterols. 

Thus, although Chol has been shown to be necessary for raft formation, the exact nature of its interaction with sphingolipids remains unclear [44]. The Chol molecule with its rigid structure has an ordering effect on the acyl chains of surrounding lipids, so its dominant role affects both the Chol-rich domains of Lo and the vast environment of the Chol-poor Ld phase. It is shown that lateral diffusion is the same for all components, regardless of their molecular structure (including Chol), if they are located in the same domain or phase in the membrane. In addition, quite unexpectedly, Chol appears to split into Ld and Lo phases to approximately the same extent, indicating that Chol does not have a strong preference for any of these phases and probably has similar interactions with all lipids. ”Reverse domains” were also observed, when strongly disordered domains with a high content of polyunsaturated FA are isolated from the Chol-rich medium [32]. It is assumed that the lateral phase separation in monolayers containing equal amounts of high- and low-Tm lipids together with Chol is due to the increasing difficulty of including an unsaturated or prenyl lipid in a highly ordered monolayer formed by saturated lipids and Chol [69]. 

## 6. Molecular Contacts between Lipids in Membranes: The Role of Sphingolipids and Polar Glycerolipids

### 6.1. Crucial Role of Sphingolipids in Membrane Construction

It is known that sphingophospholipid SM (also known as ceramide phosphorylcholine) (Figure 2B and Figure 4B), as well as Chol, are typical components of lipid rafts in animal tissue cells. Here, SM is the most widespread type of sphingolipids, accounting for 2–15% of the total number of lipid components of membranes. Formulas for SM and related phospholipids are shown in Figure 2. It can be seen that SM combines, to some extent, the properties of sphingolipids and phospholipid PC. It is believed that the intermolecular interaction of SM–SM and/or SM–Chol is responsible for dynamic molecular assemblies of lipid rafts. Hydrogen bonds and hydrophobic interactions of lipids (primarily SM and other sphingolipids) with the surrounding Chol stabilize raft-like Lo domains in membrane bilayers. However, the details of their interactions responsible for the formation of Lo domains remain largely unknown [72].

SM has similar formulas for natural compounds. These include ceramides phosphoryl-etanolamine and -serin. Hydrolysis of SM produces sphingosine, FA, phosphoric acid, and choline. Among the plant sphingoglycolipids derived from ceramides, there are cerebrosides (ceramide monohexosides), ceramide polyhexosides, and phytoglycolipids (PhGL) (Figure 4). Ceramides contain long-chain di- or trihydroxy sphingosines or phytosphingosines. In cerebrosides, the acyl groups are long-chain (up to C_26_) saturated and monounsaturated FA and long-chain 2D-hydroxy FA. It has already been mentioned that sphingolipids, Chol, and GPhLs can easily associate to form domains. The saturated acyl chains of sphingolipids allow them to be tightly packed together—a property that makes their Tm higher compared natural GPhLs, which contain more unsaturated acyl chains. Previous studies using covalent conjugates with Chol and MD simulations have shown that SM–Chol interactions are induced by a combination of intermolecular hydrogen bonds between their functional groups and van der Waals interactions of their hydrocarbon parts [41,72].

Obviously, the difference in the order of acyl chains may allow Chol to preferentially interact with SM rather than with such PC as ROPC (group 1) or DOPC (group 2), since it is known that Chol prefers ordered phospholipids [64]. Site-specific deuterated palmitoyl stearoyl PSs (PSPCs) corresponding to the acyl chain length of stearoyl SM (SSM) were synthesized, and their deuterium quadrupole bond profiles were compared in detail. The results suggest a deeper distribution of Chol in SSM membranes, a lower entropic load when placing Chol in SSM membranes, and a higher thermal stability of acyl chain orders in SSM/Chol monolayers than in PSPC/Chol monolayers at different Chol concentrations. Thus, the entropy effect and thermal stability make SM a more preferable raft component than saturated DPPC [64].

Sphingolipids generally show a preference for ordered domains. One of the functions of Chol seems to be to modulate the fluidity of sphingolipid domains, as well as to promote domain separation for functional purposes. The structure of the head groups and the composition of the acyl chains of natural sphingolipids vary greatly. Ceramide fragments with a long-chain sphingosine base and long saturated N-acyl chains contribute to the separation of sphingolipids into ordered membrane domains. The polar head group, which ranges from a single ceramide hydroxyl and SM phosphocholine group SM to large concentrations of carbohydrates in complex glycosphingolipids, will undoubtedly also affect the separation of these lipids. It is becoming increasingly clear that different sphingolipids exhibit different patterns of segregation in the lateral dimension [67].

PSM, as well as DPPC, forms the Lo phase in collaboration with Chol and has a transition temperature from the gel to the liquid phase close to that of DPPC. Unlike other phospholipids, SM is characterized by the presence of an amide group in the main chain and thus has a good configuration for forming a hydrogen bond with Chol. It is important to determine the orientation of the amide and its position in the lipid bilayer in order to understand the nature of hydrogen bonds in lipid rafts. Accordingly, the Lo phases formed by saturated GPhLs may well differ from those formed by PSM, which has implications for the length scale of Lo domains and the separation of integral membrane proteins between the two phases [15]. The tendency of sphingolipids and, in particular, SM to form hydrogen bonds is provided by the OH and NH groups at the interface of nonpolar hydrocarbon chains with polar head groups. This structural property is important both for the interaction of PSM within the bilayer assembly and for the signaling role that sphingolipid metabolites play in the cytoplasm. The OH group of PSM can act as both a hydrogen bond donor and an acceptor, while its amide group can only act as a hydrogen bond donor (Figure 2B and Figure 4B). These properties contrast markedly with the two carbonyl groups (only acceptor groups) found in the interphase region of GPhLs and play an important role in the formation of rafts through SM-Chol contacts, which is not the case for all other sphingolipids [13].

SM in biological membranes is much more saturated than naturally occurring PC. In addition, natural SMS often have a very long acyl chain linked to the amide (20–24 carbon atoms long), while the sphingoid base is most often sphingosine, which is shorter. It is sometimes assumed that such a difference in chain lengths in most naturally saturated SMS affects the curvature of the membrane and allows such SMS to participate in the inter-layer interaction of hydrocarbons [76]. However, it should be noted that sphingosine has a length comparable to conventional medium-chain FA, and some of the FA bound in GPhLs can be fatty acids with a very long chain from C_20_ to C_26_. Therefore, when comparing chain lengths, in our opinion, SM does not have any advantage over natural GPhLs.

The chemical structure of SM consists of a phosphocholine head group and a ceramide fragment consisting of hydrophobic hydrocarbon chains with polar amide and hydroxyl groups (Figure 2). Chiral amide and hydroxyl groups mainly contribute to the formation of intermolecular and intramolecular hydrogen bonds during SM interactions with other lipids and sterols. It can be noted that homophilic interactions between two SM molecules via the amide (or OH-) group seem to be the main cause of ND formation in biological membranes. It is possible that Chol interacts not so much with the polar group SM, which carries chiral centers, as with its alkyl chains [34].

Chol prefers to have more contact with SM than with DPPC. The interaction of Chol with various sphingolipids largely depends on the molecular properties of the particular sphingolipid under consideration. Thus, ceramides with their small head group (-OH) actually displace sterols from ordered domains formed by saturated PC (group 0) or SM. Other glycosphingolipids also form ordered domains in membranes, but, as in the case of DPPC, they exhibit only a small contact capacity when interacting with Chol [67]. In other words, SM and ceramides are in good contact with Chol, while cerebrosides and DPPC are worse. In order to determine the advantage of SM for the formation of the Lo phase, it is necessary to take synthetic SM with an unsaturated FA and unsaturated sphingosine base in the experiment. If the Lo phase is not formed, then it is the SM saturation that is important, and if it is formed, its head group is important. 

### 6.2. Role of Polar Glycerolipids in Membranes

The most common PLs in quantitative terms, such as PC and PE, seem to form the basis of biological membranes (matrix), but the interactions between these lipids remain insufficiently studied. PE and phosphatidylserines (PS) are aminophospholipids, i.e., they contain a primary amine in the head group. In eukaryotic membranes, PE is the most unsaturated among other lipids [76]. Figure 2 shows that the structures of SM, PC, PE, and PS molecules have a significant similarity. Hence, the functional similarity of saturated SM and DPPC in rafts and the Lo phase can be traced. For the convenience of comparing the structure of molecules, we constructed the formulas SM, PC, and PS according to the same scheme as the formula PE [77]. The lipid composition of PM varies significantly between different organisms and cell types and depends on the stage of the cell cycle, as well as on environmental factors. The average composition of mammalian PM can be represented as PC, SM glycosphingolipids, and gangliosides for the outer monolayer and PE, PS, and other charged lipids for the inner one. Mention should be made of transmembrane (transverse) asymmetry, when PL with larger polar head groups tend to be located in the outer monolayer, since there is a larger surface area per polar head group. Acyl chains of lipids can vary from fully saturated to polyunsaturated, and a large proportion of unsaturated chains belong to the inner monolayer of lipids [71]. 

Cationic lipids are often used to make artificial membranes. The head groups of uncharged lipid molecules are zwitterionic, usually with two opposite elementary charges separated by several angstroms. Common zwitterionic lipids are PC, PE, and SM. For example, the PC and PE head groups consist of negatively charged phosphate groups separated from choline or ethanolamine units that carry positively charged nitrogen. Previously, we proposed a model of the structure of the PE head group, where we found an open ring group connected by a hydrogen bond between one of the hydrogen atoms NH_3_^+^ and a negatively charged oxygen of the phosphate group (Figure 1) [77]. The phosphoethanolammonium polar head group of PE is a strong seven-membered ring structure with a neutral or weakly negative charge. The formation of a ring head group is caused by intramolecular electrostatic attraction of oppositely charged groups, as well as the presence of a hydrogen bond. The nitrogen atom in the PE head group is located at a close distance from one of the oxygen atoms of phosphate, and due to the absence of free donors/acceptors of hydrogen bonds in the ring group, it is very poorly hydrated [77]. More research is needed to understand the possible role of the PE head group structure proposed by us, as opposed to the previously known one, in the membrane structure, as well as the possibility of forming such ring head groups in other phospholipids with similar structures, such as PS.

Other common PL classes that are very important for biomembranes are charged phosphatidylglycerols (PG), diphosphatidylglycerols (DPG), PI, PS, sulfoquinosyldiacylglycerols (SQDG), and neutral high-polar phytoglycolipids (PhGL). Especially important are PI, PS, and SQDG, which are both charged and highly polar at the same time. It is likely that these three PL classes are included in both annular lipids and lipid rafts and in the Lo phase, since it is known that they (together with PhGL) tend to include increased doses of saturated FA [12].

Multivalent negatively charged lipids are also present in both synthetic and biological membranes. The head groups of these lipids can either be zwitterionic or carry an excess charge. The charge of the head groups and, consequently, the charge of the bilayers depends on the pH, the type of salt, and its concentration. The most common natural anionic lipids are PS and PG. At pH = 7, both anionic lipids have charges with charge number −1. PS has a negatively charged phosphate group, a positively charged amino group, and a negatively charged carboxyl group from the part closest to the hydrophobic group. In contrast, PG has a negatively charged phosphate group, and the terminal group (glycerol) dissociates electroneutrally. PI is a monovalent negatively charged lipid, and the carrier of this charge is the phosphate group [78]. Specific GPhLs of the inner mitochondrial membrane are DPG; they are synthesized by enzymes of the inner mitochondrial membrane and make up more than 20% of all phospholipids of this membrane. DPG increases the fluidity of the bilayer and its throughput. The presence of DPG in the bilayer leads to the formation of four-layer regions closed on all sides, similar to mitochondrial contact pads. These non-lamellar formations probably play a key role in the structure and function of membranes [12].

Saturated and unsaturated phospholipids, together with Chol, are key components of biological membranes. Saturated FA forms areas in the membrane with dense packing and regular structure. In areas with unsaturated FA, the dense packing is disrupted by bends in the hydrocarbon chain that occur due to double bonds in the cis-configuration, which prevent overly dense packing of molecules in the lipid bilayer and leads to loosening and increased fluidity of the latter. Membrane fluidity is also affected by the size of the FA hydrocarbon chains, which make the membrane more fluid as the length increases. It is known that the membrane fluidity decreases as the ambient temperature decreases. The regulation of the physical properties of membranes for temperature compensation may depend on the FA distribution at the sn-1 and sn-2 positions of glycerol in various molecular forms of some GPhSs. Thus, naturally occurring PC and PE groups contain two acyl chains, of which the chain attached to the glycerol carbon at the sn-1 position is usually saturated, while the chain esterified in sn-2 is mostly unsaturated. An unsaturated acyl chain can contain from one to six double bonds of the cis-configuration [12,76].

Polyunsaturated phospholipids interact comprehensively with other components of the membrane. Paired interactions between individual lipids in the three-component lipid monolayer DPPC (group 0)/palmitoyl linolenoyl-PC (PLePC, group 1)/Chol, forming the Lo and Ld phases, were studied using MD simulations using all atoms. The results showed that PLePC exhibits strong repulsion with DPPC and Chol in the Ld phase. When the bilayer passes into the Lo phase due to an increase in the concentration of Chol, the repulsion of PLePC with DPPC and Chol is significantly reduced. As a rule, phospholipids containing polyene FA play a strong repulsive role in the Lo phase and are found in the Ld phase. Thus, the interactions between saturated lipids and Chol are attractive, whereas the contacts Chol has with unsaturated lipids are repulsive due to the double bonds in their acyl chains. 

It is generally accepted that the rafts or Lo phase in PM owe their existence to a combination of sphingolipids (primarily SM) and Chol. It probably is. What about GPhLs, as opposed to sphingophospholipids, which are SM? It is known that GPhLs, along with Chol and SM, serve as necessary components of PM in both animal and plant tissues, and are components of domains, rafts, and the Lo phase. A number of papers mention that the rafts or Lo phase may contain, in addition to Chol and sphingolipids, GPhLs [6,14,15,65]. Very often, the authors discuss the role of Chol and SM in the formation of rafts or the Lo phase by excluding one or the other from the studied ternary mixture and thereby showing the degree of their necessity [32,37,44,65,67]. Such experiments usually include only one of the different saturation GPhLs (most often PC groups 0, 1, and 2). However, conclusions are often drawn only about the benefits of Chol and SM, and the role of these phospholipids in triple mixtures is rarely discussed. It would be important to conduct an experiment with the inclusion or, conversely, exclusion of saturated GPhLs in the model experiment with a triple mixture of PC/Chol/SM and clearly show the role of these GPhLs in membrane rafts (Table 1). It is probably just as important as the role of Chol and SM. Is it possible that Chol and SM alone (without saturated GPhLs) can form rafts? They probably can if there is an Ld phase with a mass of h-unsaturated phospholipids around. The presence of saturated GPhLs in rafts is likely to increase their workability. A similar role for palmitic acid in saturated GPhLs is naturally played by both myristic and stearic FA, as well as saturated FA with a very long chain—C_20_, C_22_, C_24_, and C_2_. Unsaturated GPhLs are the main components of the Ld phase. Finally, half-saturated GPhLs (group 1) can enter both of the Lo and Ld phases and, in addition, can serve as an intermediate layer between the Lo and Ld phases (Figure 1D) [51].

When analyzing rafts of the PM outer monolayer or modeling these rafts, it is often assumed that the composition of the three raft components is SM/POPC/Chol [47,79]. This minimal system simulates the composition of PM and reproduces many complex phenomena, including the coexistence of the Lo and Ld phases; however, their compositional simplicity makes it suitable for studying composition- and temperature-dependent behavior [13]. Variants in this triple group for PSM are DPPC [13,64] or DSPC [46,47,79] (Table 1). SM, DPPC, and DSPC are highly fusible lipids capable of forming a gel phase [62]. Instead of POPC (group 1) in model mixtures, we can also use DOPC (Group 2) [13,28,41], which will further reduce the Tm value in the domain. We could also use a lipid of the same group 1, not with monoenic FAs but with diene (linoleic) [74], triene (linolenic; PLePC), or even polyene (docosahexaenoic) FAs (PDPC) [51]. PDPC—palmitoyldocosahexanoyl-PC is a group 1 GPhL but enhanced by polyene FA 22:6^Δ4,7,10,13,16,19^. Amphiphilic PC in parallel experiments can be replaced by highly charged DPG, as well as PI, which is both a charged and highly polar annual lipid, or by a glycolipid (for example, monogalactosyldiacylglycerol). At the same time, we should not forget that any of these types of lipids can include three groups with different unsaturation, as well as PC (see above).

In experiments with models of other membranes, a number of other sterols—precursors of Chol—lanosterol, sitosterol, stigmasterol, or campesterol, can replace Chol. SM can be substituted with ceramides or glycosphingolipids (cerebrosides, ceramide polyhexosides, or phytoglycolipids (PhGLs)). Additionally, if it is used in the PC modeling scheme, then in parallel, it is necessary to include experiments with Chol in seven of its three groups, with DPPC, DOPC, or POPS, as well as with the absence of PC, and compare the results obtained with each other (Table 1). Moreover, in experiments with the inclusion of DPPC, it is important to either exclude SM or include it in parallel and compare the effect of two saturated phospholipids together or just one of them. It will be interesting to confirm that unsaturated PC (DOPC) will not enter the raft and also to see in which zone of the raft (or not of the raft) the POPC will be detected; it will probably enter both zones or it will be located at the interface of the two phases, in the pro-daily zone. It is interesting what happens if you take into account all three PCs and Chol, but without SM. It can be assumed that they alone can cope with the formation of phases Lo and Ld [46]. Additionally, if you take SM in the same experience but do not take Chol, then no phases are formed. What about without Chol? Or maybe even in this case, the phases are formed [33,72]. We present a sample scheme for setting up simulation experiments, which takes into account both the two- and three-component mixtures used earlier and the four-component mixture proposed by us, which is closer to the composition of lipids in natural membranes (Table 1).

The important role of low Tm lipids (e.g., DOPC or POPC) in mesoscale control of the size of the membrane domain can be noted. DOPC is not often detected in cell membranes but is popular in model membrane experiments because of its propensity to induce micron-sized liquid domains in ternary mixtures. In contrast, POPC is a major lipid with low Tm in the outer flap PM. Triple mixtures containing lipids with high Tm (SM or DPPC)/Chol/POPC appear to exhibit exclusively nanoscale Ld + Lo heterogeneity. At the same time, micron-sized domains were formed in mixtures containing DOPC under similar conditions [13]. 

Theoretically, the Tm value should vary in the DPPC–POPC–DOPC or PLPC–PLePC–PDPC series from high to low values, from a high-melting (with a large Tm) to a low-melting lipid with a smaller Tm. PM membranes obviously involve not only PC but also PE; therefore, in modeling, POPC was sometimes used along with POPE and along with DOPC–DOPE, since PE is more often unsaturated in nature than PC. POPC would be better to replace palmitoyl linoleoyl PC (PLPC) due to its greater prevalence in nature, linoleic rather than oleic FAs. 

SM had a favorable effect on the formation of ultra-ND domains compared to DPPC; the formation of large domains was less affected by SM. Natural mixtures of mammalian lipids are probably more prone to the formation of ultra-ND than large domains, and the domain size is more sensitive to the membrane composition than the domain formation itself [46].

## 7. Conclusions

Biological membranes are difficult to study in terms of their molecular composition and structure; they function over a wide range of time scales and are characterized by nonequilibrium conditions. Therefore, modeling is one of the most appropriate methods for studying the behavior of biomembranes. Lipid rafts are increasingly being confirmed in membranology, and we can expect that similar structures will be found, in addition to PM and tonoplast, in a number of membranes of various organelles. These lipid rafts, or rather “icebergs,” penetrate each monolayer of the membrane to its full depth and not as it was originally thought in the case of GPI proteins. In terms of their size, rafts can be of two types—2–20 nm (ND) and 20–200 nm (domains of lipid rafts). Lipid rafts and the Lo phase, if not the same, are very close to each other in composition and structure. The Lo phase may consist of Chol, sphingophospholipids (SM), sphingoglycolipids (cerebrosides and others), and GPhL groups 0 and 1. The suggested composition of the Ld phase is GPhL groups 2 and 1, Chol, and unsaturated glycolipids. Based on the above, we can conclude that the minimum unit of the membrane consists of seven molecules, which includes one DPPC molecule occupying a place in the center and two Chol, POPC, and DOPC molecules located on each side of the DPPC. Further, these units can either merge with each other or alternate in the membrane. Each of these connections can be replaced by others (see above).

Instead of discussing the role of “saturated” and “unsaturated” phospholipids in membranes, we proposed dividing each of the GPhLs, as well as glycolipids, into three groups according to the degree of unsaturation (groups 0, 1, and 2), depending on the composition of their FA residues. It seems that the three-component mixture of lipids used by many authors for modeling the formation of Lo and Ld phases and lipid rafts does not provide reliable information about the composition of the Lo and Ld phases. The main result obtained with its use is the justification of the necessity of SM and Chol for the formation of rafts and Lo and Ld phases in PM models. We have proposed a scheme for setting up experiments that includes a four-component mixture; this scheme allows us to obtain more reliable information about the composition of the Lo and Ld phases. The fact that the model membrane contains an intermediate zone between the Lo and Ld phases, as well as the formation of denser formations in the Lo phase, indicates that the boundaries between the Lo and Ld phases may be blurred and not as clear as is usually assumed. It can be assumed that the membranes do not have a block but a wave character of their structure, in which the Ld, ND phase, and microdomains of lipid rafts alternate in their composition, not forming clear boundaries but smoothly changing this composition at their boundaries.

The importance of the head groups of polar lipids and sterols in membranes is unmistakable, and this can be seen in the example of Chol; if you take away this group and add a hydrophobic chain instead, the resulting Chol esters may also be unnecessary for the membrane. A configuration of the PE molecule is proposed, taking into account the possibility of the existence of a seven-membered ring head group in it.

The configuration (geometry) of molecules plays a crucial role in the formation of membranes. Thus, if the PC and PE molecules, as well as the SM (ceramide phosphorylcholine) molecule, are equally successfully used in eukaryotic membranes, then the ceramide phosphoryl ethanolamine molecule for some reason does not find application in these membranes and is found only in bacteria. Molecules of a number of compounds, such as Chol and SM, play a crucial role in membrane biogenesis of Chol and SM. The properties of Chol required for membranes are lost in lanosterol or in Chol esters (see above). SM acts with success in animal tissue cells but has not found application in plant cells, and Cer, on the contrary, is actively involved in the biogenesis of plant cell membranes, but in animal tissues, it is only slightly present in the form of gangliosides. It would be important to build 3D models of the molecules that make up the membrane, primarily Chol, SM, cerebrosides, DPPC, DOPC, POPC, and their PE-based analogues, and combine their shapes with each other in the way they are presumably located in the rafts and phases of Lo and Ld. It is necessary to study the structure of these molecules and create more and more new models of membranes.

## Figures and Tables

**Figure 1 ijms-25-08325-f001:**
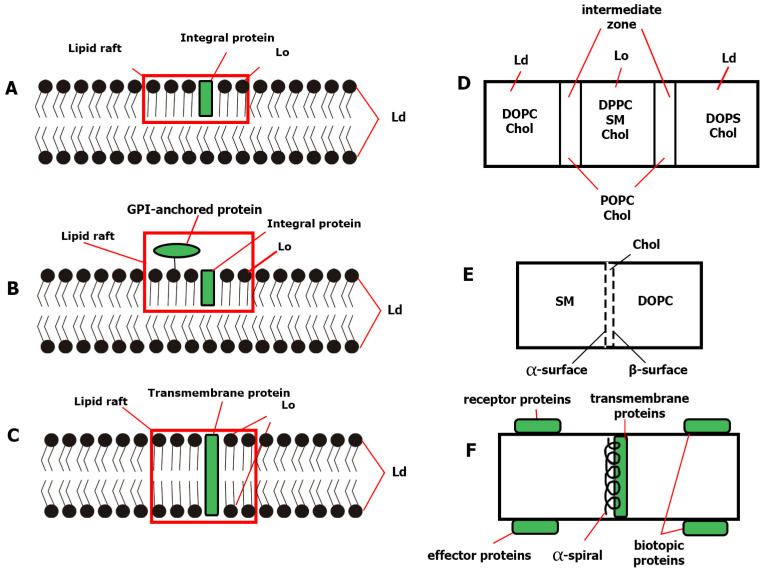
Models of the possible organization of lipid rafts in eukaryotic cells. Some of the rafts in 1–3 in the membranes can take place even without the proteins included in them. (**A**)—location of the raft in one of the membrane monomers. (**B**)—location of the raft including the GPU in one of the membrane monomers. (**C**)—a raft occupying part of both monolayers of the membrane. (**D**)—possible intermediate zones between the phases of Ld and Lo. (**E**)—α- and β-surfaces of Chol, reversed by SM and POPC, respectively [28]. (**F**)–location of receptor, effector and biotopic proteins in the membrane. Ld—liquid disordered lipid phase, Lo—liquid ordered lipid phase, DOPC—dioleoyl phosphatidylcholines, DPPC—dipalmitoyl phosphatidylcholines, SM—sphingomyelin, POPC—palmitoleoyl phosphatidylcholines, α-spiral—a protein molecule with an α-spiral structure, Cer—silversides, Chol—cholesterol, α- and β-surface—two surfaces of the Chol molecule.

**Figure 2 ijms-25-08325-f002:**
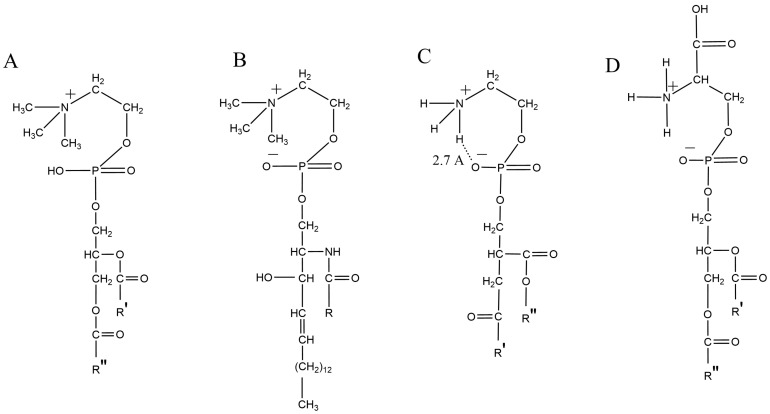
Chemical forms of certain types of phospholipids and sphingophospholipids involved in the construction of lipid rafts. (**A**)—phosphatidylcholine (PC), (**B**)—sphingomyelin (SM), (**C**)—phosphatidylethanolamine (PE), (**D**)—phosphatidylserine (PS).

**Figure 3 ijms-25-08325-f003:**
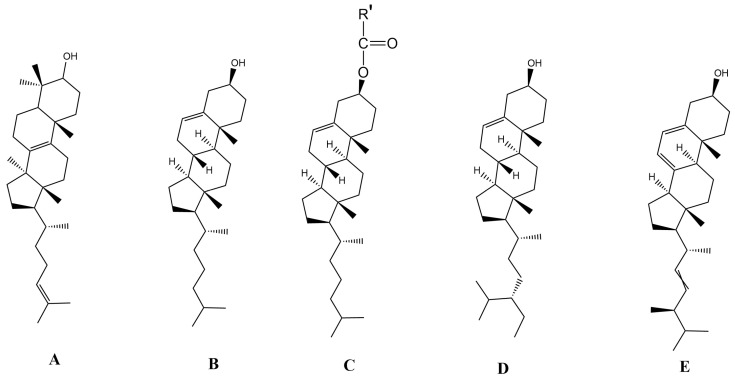
Chemical formulas of sterols involved in the construction of rafts, their synthesis precursors, and products of further transformation. (**A**)—lanosterol, (**B**)—Chol, (**C**)—esters of Chol and FA, (**D**)—β-sitosterol, (**E**)—ergosterol.

**Figure 4 ijms-25-08325-f004:**
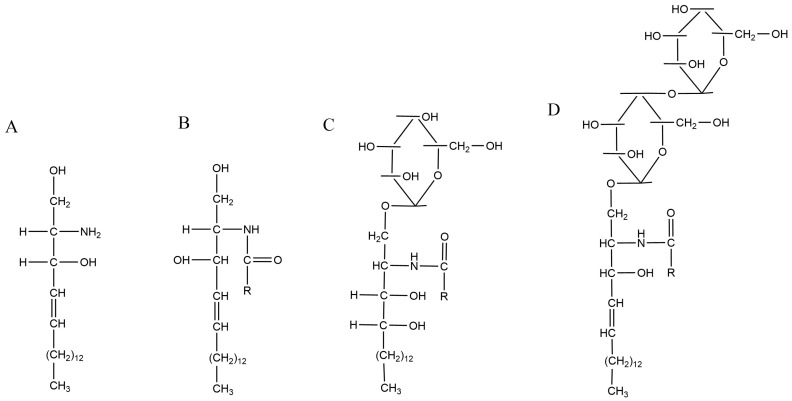
Chemical formulas of sphingosine, ceramides, and certain types of sphingoglycolipids involved in the construction of lipid rafts. (**A**)—sphingosine, (**B**)—ceramides, (**C**)—cerebrosides, (**D**)—dihexosides of ceramides.

**Table 1 ijms-25-08325-t001:** A possible scheme for setting up experiments on modeling the formation of rafts or Lo and Ld phases with different compositions of lipid mixtures. Chol—cholesterol, PSM—palmitoyl sphingomyelin, DPPC—dipalmitoyl phosphatidylcholines, POPC—palmitoleoyl phosphatidylcholines, DOPC—dioleoyl phosphatidylcholines, Cer—cerebrosides, GPhLs—glycerophospholipids, SM—sphingomyelin.

№	1	2	3	4	5	6	7	8	9	10	11	12
Chol	+	+	+	+	+	+	+	+	+	+	+	+
PSM	+	+	+	+				+	+			
DPPC	+				+		+	+		+		
POPC		+				+			+		+	
DOPC			+			+	+	+	+			+
Cer										+	+	+
References	[13,64,65,66]	[13,14,15,47,48,65]	[13,14,15,28,37]	[67]	[67]	[13]	[68]	Prop-osed	Prop-osed	Prop-osed	[51]	Prop-osed

Notes: PSM, two-classical scheme, DPPC, POPC, and DOPC can be replaced by any GPhL of groups 0, 1, and 2, respectively; Cer can be replaced by dihexosylceramides; unsaturated SM can be used instead of unsaturated PSM.

## Data Availability

The data are contained within the article.

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
