# Peer review of "Polar Glycerolipids and Membrane Lipid Rafts"

_ijms, 2024, doi:10.3390/ijms25158325_

Round 1

Reviewer 1 Report

Comments and Suggestions for Authors

I cannot agree with this paper from the very beginning. The first sentence in the abstract reads: "Current understanding of the structure and functioning of biomembranes is impossible without determining the mechanism of formation of membrane lipid rafts." Well, my point of view is the opposite: the raft misconcept has muddled  and confused the whole field of biomembrane structure. The authors' view joins the majority opinion on this matter, against the physical chemical evidence, and thet fail to cite a single paper of those that represent the anti-raft view.

I would advise the authors to read these two reviews, and then re-write the manuscript, although I doubt that they will undertake such an endeavor. In any case, the reviews are: https://doi.org/10.1016/j.chemphyslip.2018.11.006 and DOI: 10.1002/9780470015902.a0028879.

Author Response

Dear reviewer, thank you so much for carefully reading our review article. In your review, you acknowledge that we join to most authors on the existence of such a term as “membrane lipid rafts". In our work, we did not set out to understand terminology issues. Moreover, we quite agree with your opinion that it is probably better to write not "membrane lipid rafts", but "membrane nanodomains". However, in order not to increase the volume of the text by justifying such a replacement, we, as already mentioned, joined the opinion of the majority of the authors. One of the articles mentioned by you and devoted to nanodomains is already cited by us under the link [50]. In this work, we are really talking about nanodomains. And in our work, we are also talking about domains of different sizes, including nanodomains, and membrane lipid rafts. We agree that both of these names have a place to be, but our main task is to discuss the role of various polar glycerophospholipids involved in the structure and functioning of both nanodomains and lipid rafts.

Reviewer 2 Report

Comments and Suggestions for Authors

This review summarizes the roles of glycerophospholipids (GPhL), sterols (especially Chol), and sphingolipids (especially SM) in membrane phase separation. Although the text is verbose, the manuscript is based on many references and focuses on the saturation of lipids, and it deserves to be published. It should be published with some modifications.

1.     Since each chapter is long, dividing it into subsections with headings would enhance readability.

2.     Table 1 lists lipid compositions used in research, but lacks information on their specific properties discussed in the text, such as phase separation, advantages & disadvantages as model membranes, which should be included along with references.

3.     Figure 3: Since the text describes the difference in structure between the α-plane and the back surface of cholesterol, the stereochemistry of the methyl groups should be shown correctly in Figure 3.

4.     L749: "But in plant sterols, which are also very important in building PM, the head groups are not small. So, in sitosterol, the head group is a fairly large galactose group." In plants, the major sterol is β-sitosterol (Fig. 3-4), which has no sugar.

5.     Definitions are needed for several terms: "aflesh-like domain" (L337), "flesh-like membrane" (L704), "aktina-based membrane" (L382).

6.     Add explanation to Figure 1 caption. Where the text refers to Figure 1 (L251, L349, L421, L643, L645, L1042), specify which figure it is, as in Figure 1-1.

7.    Abbreviations are treated incorrectly in some places (no formal name at the beginning or full spelling after the second time. Also, capitalization is often incorrect and should  be checked carefully.

8.     L658: "phosphatidylinosites" should be corrected to phosphatidylinositol?

9.     L685: Since the structure of cholesterol is discussed, should Figure 2 be referenced?

10.  L887: Figure 4 is unrelated to the text's explanation.

11.  L903: Since it discusses sphingolipids, shouldn't it refer to Figure 4?

Comments on the Quality of English Language

Abbreviations are treated incorrectly in some places (no formal name at the beginning or full spelling after the second time. Also, capitalization is often incorrect and should  be checked carefully.

Author Response

We express our deep gratitude to the reviewer for the careful reading of our manuscript and the comments made on the text.

  1. Since each chapter is long, dividing it into subsections with headings would enhance readability.

Answer: We have divided a number of chapters into subsections

  1. Table 1 lists lipid compositions used in research, but lacks information on their specific properties discussed in the text, such as phase separation, advantages & disadvantages as model membranes, which should be included along with references.

Answer: The properties of each mixture are described in sufficient detail in the text of the article. links at the request of the reviewer to each mix variant indicated in the table provided links

  1. Figure 3: Since the text describes the difference in structure between the α-plane and the back surface of cholesterol, the stereochemistry of the methyl groups should be shown correctly in Figure 3.

Answer: We have provided a more detailed picture of cholesterol, as well as other sterols

  1. L749: "But in plant sterols, which are also very important in building PM, the head groups are not small. So, in sitosterol, the head group is a fairly large galactose group." In plants, the major sterol is β-sitosterol (Fig. 3-4), which has no sugar.

Answer: The text has been changed, taking into account the comments of the reviewer

  1. Definitions are needed for several terms: "aflesh-like domain" (L337), "flesh-like membrane" (L704), "aktina-based membrane" (L382).

Answer: The text has been changed, taking into account the comments of the reviewer. "aflesh-like domain" (L337), "flesh-like membrane" (L704) was replaced with "raft-like domain" and "raft like membrane"

  1. Add explanation to Figure 1 caption. Where the text refers to Figure 1 (L251, L349, L421, L643, L645, L1042), specify which figure it is, as in Figure 1-1.

Answer: Done

  1.   Abbreviations are treated incorrectly in some places (no formal name at the beginning or full spelling after the second time. Also, capitalization is often incorrect and should  be checked carefully.

Answer: Thanks again for the careful reading of the text, the translation errors have been corrected

  1. L658: "phosphatidylinosites" should be corrected to phosphatidylinositol?

Answer: Done

  1. L685: Since the structure of cholesterol is discussed, should Figure 2 be referenced?

Answer: Done

  1. L887: Figure 4 is unrelated to the text's explanation.

Answer: Added a mention in the text

  1. L903: Since it discusses sphingolipids, shouldn't it refer to Figure 4?

Answer:  We referred to Figures 2.2 and 4.2. Figure 2.2 shows the formula of sphingomyelin, and Figure 4.2 shows the formula of ceramide, which is part of sphingomyelin

Comments on the Quality of English Language

Abbreviations are treated incorrectly in some places (no formal name at the beginning or full spelling after the second time. Also, capitalization is often incorrect and should  be checked carefully.

Answer: Done

Round 2

Reviewer 1 Report

Comments and Suggestions for Authors

I cannot change my views, and I cannot expect that the authors will change theirs.